# Thompson Sampling for Bandits with Clustered Arms

## Reproducibility Summary

**Scope of Reproducibility**

This report covers our reproduction of the paper 'Thompson Sampling for Bandits with Clustered Arms' by Carlsson et al. (IJCAI 2021) [1]. The authors propose a new set of algorithms for the stochastic multi-armed bandit problem (and its contextual variant with linear expected rewards) in settings when the arms are clustered. They show both theoretically and empirically that exploiting the cluster structure significantly improves the obtained regret over the traditional assumption with non-clustered arms. Furthermore, they compare the proposed algorithms to previously proposed and well-known benchmarks for the bandit problem. We aim to reproduce just the empirical evaluations.

**Methodology**

Given that no code was provided alongside the original paper (and neither for any of the benchmarks used for comparison), we implement everything from scratch. We write our code in R, the well-known programming language for statistical computing, and use some of its basic libraries. We run the experiments on a laptop with a dual-core Intel i7 processor and 8 GB of RAM. We don't use any GPU.

**Results**

There are no exact numbers in the original paper to reproduce, rather the main claims are supported with some visualisations. With this in mind, our reproduction confirms the advantage provided by clustering over the assumption of independent arms, as well as the newly proposed algorithms outperforming the referenced benchmarks. We repeat all the experiments with multiple seeds to obtain robust estimates of the algorithms' performance and reduce the risk of drawing any conclusions out of results obtained by chance.

**What was easy**

The authors have included in the paper all the necessary details to reimplement their proposed algorithms, recreate the synthetic datasets, and reproduce the experiments for the first part, i.e. the traditional multi-armed bandits setting.

**What was difficult**

It is much harder to reimplement some of the referenced benchmarks. The main reasons for these struggles are the inconsistent nomenclature, important details missing in the referenced papers, and some of the compared benchmarks being originally designed to run in a different setting. Furthermore, some additional research into the field of contextual bandits is needed to reproduce the second part of the experiments.

**Communication with original authors**

There has been no communication neither with the authors of the original article nor with any authors of the referenced papers.

# 1 Introduction

The multi-armed bandit is a classical reinforcement learning problem, formally defined for the first time already by Robbins in 1952 [2], focused on the exploration versus exploitation trade-off. Its name is derived from gambling: imagine a gambler sitting in front of a row of slot machines, deciding which arm to pull next hoping to maximise the prize money (to minimise his loss, actually). The same name is used for whatever problem involves a learner and a fixed set of $N$ actions to repeatedly choose from, with each action returning a stochastic reward. The learner aims to maximise its obtained reward in a finite number of steps $T$. He does therefore repeatedly face the dilemma of whether to exploit what he believes to be the most convenient action, or to explore other actions, hoping to find an even better one. In the beginning the learner might have been provided with some additional info about the available actions or not, it depends on the setting of the problem. Anyway, he constantly updates his knowledge about the actions based on the obtained rewards.

An interesting and useful generalisation is the contextual bandit. At each iteration the learner gets an additional context vector which he can use, in addition to the past rewards, to choose his next action. His goal over time is to get enough information about how the context vectors and rewards are related, to be able to predict the best arm by looking at feature vectors.

Both mentioned types of multi-armed bandits are analysed also in the paper we aim to reproduce, *Thompson Sampling for Bandits with Clustered Arms* by Carlsson et al. [1]. The problems defined in the article have an additional, defining characteristic: the arms are clustered (both in the analysed classical and contextual bandits). The leading idea is to understand how much can the obtained reward be optimised if the learner can leverage the additional knowledge on the relations between arms.

The multi-armed bandit problem (with its multiple variants) can be successfully applied to some common usecases. Let's take a recommender system on an e-commerce website as an example to better explain all the mentioned settings. When a user visits the homepage of the website, an agent in the background has to decide which product to show him first, hoping he will eventually buy it (thus obtaining a reward). Without any additional knowledge about the user, this example represents a classical multi-armed bandit. However, the items on sale are not completely unrelated between them. If they can be grouped in meaningful clusters, for which we expect similar selling success, the problem translates to a multi-armed bandit with clustered arms. These clusters might even be hierarchically structured (e.g. the items are clustered into *electronics* and *clothes*, with the latter further divided into *sports* and *elegant clothing*). Furthermore, if some data about the user is also available (i.e. a *context vector*), we talk about a contextual multi-armed problem, with the context influencing the obtained reward (since users' preferences vary).

# 2 Scope of reproducibility

The main goal of the original paper is *"to show, both theoretically and empirically, how exploiting a given cluster structure can significantly improve the regret and computational cost compared to using standard Thompson sampling"* [1]. To achieve it, new algorithms based on a multi-level Thompson sampling scheme [3] are proposed. These algorithms are designed to solve the stochastic multi-armed bandit with clustered arms (MABC) problem and its contextual variant (CBC) with linear expected rewards (linearly dependent on the given context vector). Under some specific assumptions (mainly strong dominance of the best cluster and Bernoulli distributed arm rewards), theoretical bounds on the regret are provided, with the dependence on the number of arms $N$ removed in favor of dependence on the properties of the selected clustering. The algorithms are then tested on some specifically constructed datasets that meet the assumptions, as well as compared with other recently proposed algorithms that solve the MABC and CBC problems even in settings where the theoretical assumptions are violated. The results indicate the theoretical guarantees hold true and the proposed algorithms are at least comparable with the evaluated baselines (and most often outperforming them).

In this work, we aim to fully reproduce all the experiments described in the paper. We divide the claims we focus our work on into three subsections for clearness.

1) Classical MABC setting with clusterings that meet the required theoretical assumptions
   - In flat clusterings, the proposed algorithm (TSC) outperforms the baseline (Thompson sampling – TS).
   - In hierarchical clusterings, the proposed algorithm (HTS) outperforms the baseline (TS).
   - Regret does only depend on the clustering quality, not on the number of arms $N$.
2) Classical MABC setting with clustering that violates the defined assumptions
   - The proposed algorithms (TSC, HTS) still perform better than TS (in both flat and hierarchical settings).

- TSC and HTS are at least comparable (and most often better) than other, recently proposed algorithms that also solve the MABC problem.

3) Contextual bandits variant (CBC)
- The proposed algorithm LinTSC outperforms the baseline LinTS.
- LinTSC is at least comparable and most often better than other, recently proposed algorithms that also solve the CBC problem.

# 3 Methodology

Given that no code was provided alongside the original paper (and neither with any of the articles where the benchmarks used for comparison are described), we implement everything from scratch [1]. Additional details about the proposed algorithms and how we implement them follow in the next section 4.

## 3.1 Datasets

We don't use any externally provided dataset, all the experiments are run with synthetically generated data, following the instructions in the paper. For each experiment we prepare a separate dataset. The multi-armed bandit is a special case of a reinforcement learning problem, and as such doesn't require any split into training or test data, the agents learn and take actions simultaneously.

## 3.2 Computational requirements

We write our code exclusively in R, the well-known programming language for statistical computing, and use some of its basic libraries for data manipulation and visualisation (e.g. *ggplot2*). We run the experiments on a laptop with a dual-core Intel i7 6th generation processor and 8 GB of RAM. We don't use any GPU. The computational resources are very limited, but the nature of the problem doesn't require huge processing, thus allowing us to smoothly run all the required experiments. Some slow down is observed in the runs with a higher number of arms and actions, but we still manage to evaluate the models over multiple (up to 100) seeds in a couple of hours, which is crucial to get accurate estimations of their performance.

# 4 Experiments

As described in [1], a standard multi-armed bandit (MAB) problem is defined with a set of $N$ arms $\mathbb{A}$, a finite number of steps $T$ and reward functions $r_t(a_t)$ which depend on the played arm $a_t$ at the timestep $t$, but might as well depend on the timestep $t$ itself. In MABC and CBC problems, the arms are additionally divided into (flat/disjoint or hierarchical) clusters. Rewards are drawn from some distribution $r_t \sim \mathbb{D}_{a_t}$, with an unknown mean $\mathbb{E}_{D_{a_t}}[r_t] = \mu_{a_t}$. The goal of the learner is to maximise the expected cumulative reward over a sequence of $T$ time steps or, equivalently, to minimise its expected cumulative regret $\mathbb{E}[R_t]$ w.r.t. the optimal arm $a_t* = argmax_{a_t \in \mathbb{A}} \mu_{a_t} \forall t \leq T$ (the cumulative regret represents how much reward did the learner lose due to not always playing the best arm [2]). Rewards might be drawn from arbitrarily chosen distributions, but to simplify the derivation and proof of theoretical bounds for the cumulative regret, only Bernoulli and uniformly distributed rewards are used in the original paper. Since the reward functions differ in different MAB settings, we provide additional details about them in the following sections.

## 4.1 Classical MABC

In the experiments with classical MABC problems, all the rewards are drawn from a Bernoulli distribution $r_t(a_t) \sim Bernoulli(\theta_{a_t})$. The parameters $\theta_a$ are defined in advance (but not known to the learner) and constant for each arm $a$, therefore the cumulative regret can be defined as $R = \sum_{t=1}^{T} \theta_{a^*} - r_t(a_t)$, where $\theta_{a^*} = max_{a \in \mathbb{A}} \theta_a$ is the expected reward for playing the best arm. The arms are divided into clusters based on their $\theta$ value. These clusters might be disjoint (each arm gets assigned to exactly one cluster) or hierarchical (each arm gets assigned to exactly one leaf in a clustering tree). Not all algorithms can solve both types of the MABC problem.

The baseline algorithm that solves a MAB problem is the Thompson sampling (TS) [3], first designed by Thompson in 1933 (much before the MAB problem was even formalised). It doesn't take into account any clustering information

---

[1] The code required to reproduce all the experiments is available on the GitHub repository `https://anonymous.4open.science/r/reproducibility_challenge-CEA3`

[2] Notice that the regret might be (and often is) negative at single timesteps when positive rewards are observed.

(thus being able to solve all the proposed MABC settings). The main idea behind it is to select which arm to play at the current step probabilistically, i.e. with respect to the current belief about the arms reward distributions. The learner starts with assigning uninformative $Beta(1,1)$ priors over expected rewards $\theta_a \in [0, 1]$ to each arm $a$ (the $Beta$ distribution was chosen due to its conjugate characteristic). Then, at each step, it takes a random sample from the $Beta(S_t(a), F_t(a))$ distribution for each arm, and plays (greedily) the arm with the highest sampled expected reward. The posterior belief in that arm's true $\theta$ value is then updated according to the observed reward $r$: $S_{t+1}(a) = S_t(a) + r$, $F_{t+1}(a) = F_t(a) + (1 - r)$. Since the learner doesn't get any information about the arms that were not played, the other posteriors are not updated.

The newly proposed algorithm that exploits the disjoint clustering structure (TSC [1]) is based on the exact same idea as TS, but adds an additional level to the selection of the arm to play. Instead of sampling from the arms' priors directly, it keeps prior beliefs for the clusters too and samples from them first. When the cluster with the highest sampled expected reward is selected using TS, the procedure is repeated for the arms within that cluster. Then the posteriors (both for the selected cluster and played arm) are updated according to the observed reward.

The proposed algorithm for the MABC problem with hierarchically clustered arms (HTS [1]) is a natural extension of TSC: it applies the Thompson sampling at each node to select in which subtree to search for the arm to play. TSC is basically a HTS on a tree with depth 2, while TS is a HTS on a tree with depth 1 (a single node with $N$ leaves).

The assumptions on the clustering structures required for the theoretical regret bounds to hold are quite tight, assuming strong dominance (and hierarchical strong dominance) between the clusters. This actually means that every arm from the optimal cluster must have a higher expected reward than any other arm from the other clusters (in the hierarchical structure, this condition is applied at each node level). The authors, therefore, provide precise instructions for the construction of synthetic datasets on which the algorithms are tested.

To build a strongly dominant disjoint clustering structure on which to test the proposed algorithms, we need to define the following hyperparameters:

- the number of arms $N$,
- the number of clusters $K$,
- the size of the optimal cluster $A^*$,
- the width of the optimal cluster $w^*$,
- the distance of the other clusters to the optimal one $d$,
- and the number of timesteps $T$.

The arms are divided into clusters randomly. The probabilities assigned to the arms in the optimal cluster are sampled uniformly from $U(0.6 - w^*, \ 0.6)$, while those in the other clusters are sampled from $U(0.5 - w^* - d, \ 0.6 - w^* - d)$. In all the clusters, two arms get assigned the upper and lower bound of the interval their values were sampled from (e.g. the highest expected reward is always 0.6). In the Results section 5, we show how those hyperparameters influence the obtained cumulative regret.

The hierarchical datasets are built in a completely different way (and require fewer hyperparameters). The probabilities assigned to the arms are sampled uniformly from $U(0.1, \ 0.8)$ and then recursively sorted and merged into a balanced binary tree that meets the hierarchical strong dominance assumptions (i.e. at each node, the top half of the arms gets assigned to a subnode and the bottom half to the other). Other than the number of arms $N$ and timesteps $T$, the defining hyperparameter is the number of levels $L$ [3].

### 4.1.1 Classical MABC with violated assumptions

The strong dominance condition is difficult to meet in real-life scenarios, therefore we evaluate the performance of the proposed algorithm also on datasets where this assumption is not met, and compare them to other well-known algorithms that solve the MABC problem. We generate the synthetic datasets in a completely different way: we assign a parameter $x_a \sim U(0, \ 1)$ to each arm, group them into $K$ clusters using the K-means algorithm and then convert their parameters to probabilities with $\theta_a = f(x_a)$, where $f(x) = \frac{1}{2}(\sin{(13x)}\sin{(27x)} + 1)$. The function is smooth, therefore arms in the same cluster have similar expected rewards, but its periodicity ensures there are no strongly dominant clusters. For the hierarchical structure, we repeat the same process at each level.

The proposed algorithms based on Thompson sampling are compared to the following ones:

---

[3]Notice that a single-level tree represents a MAB, and a two-level one a MABC problem.

- The UCB1 [4] (Upper Confidence Bound) algorithm solves the MAB problem (ignores the clustering structure). It is based on a deterministic policy, which at each step selects the arm that maximises the expression $\bar{r_a} + c_p\sqrt{\frac{2lnn}{n_a}}$, where $\bar{r_a}$ is the average reward obtained from arm $a$ so far, $n_a$ the number of times $a$ was played and $n$ the total number of plays. Each arm needs to be played once at the beginning for initialisation.

- The UCBC algorithm (designed as TLP – Two Level Policy – by Pandey et al. [5], named UCBC in [6]) is an extension of UCB1 to clusters. It uses a two-level selection schema, with first selecting the best cluster with respect to the UCB1 formula, and then playing the best arm from the cluster. A policy on how to represent the clusters need to be chosen. Since the authors of [1] don't mention which one they use, we choose to implement the MAX policy (which [5] states to perform best). With the MAX policy, each cluster is represented by its best arm (other proposed policies are MEAN and PMAX).

- The TSMax algorithm (named HTS when first proposed in [7], renamed to avoid misunderstandings) is extremely similar to the TSC. The only difference is that the clusters' posterior beliefs are defined as the posterior of the current best arm inside the cluster.

- The UCT algorithm (Upper Confidence Bound for Trees [8]) is an extension of the UCB1 algorithm to hierarchical clustering structures. It applies the UCB1 selection procedure recursively at each node and selects the most promising one until a single arm is selected.

## 4.2 Contextual CBC

In the experiments with CBC problems, the expected values of the rewards are linearly dependent on the context vector $x_t \in \mathbb{R}^d$ and arm parameters $\theta_a \in \mathbb{R}^d$: $\mathbb{E}[r_t(a)|x_t] = x_t^T\theta_a$. The parameters $\theta_a$ are defined in advance (but not known to the learner) and remain constant throughout the experiment, while a different context $x_t$ is observed at each timestep $t \leq T$. The rewards are uniformly distributed $r_t(a_t) \sim U(0, 2x_t^T\theta_{a_t})$. The cumulative regret is defined as $\sum_{t=1}^{T} x_t^T\theta_{a_t^*} - r_t(a_t)$, where $\theta_{a_t^*} = argmax_{a \in A} \ x_t^T\theta_a$ is the best arm for the given context (the best arm is not always the same). The arms are grouped into clusters based on their $\theta_a$ parameter vector. We use only disjoint clusterings in our experiments.

A baseline algorithm, derived from Thompson sampling, that solves the CBC with linear expected rewards is the LinTS (first mentioned by Agrawal et al. [9]). It's similar to TS in the MABC setting: at every step it samples from the prior distributions of the arms' parameters, plays the arm with the highest sampled expected reward and it doesn't use any clustering information. The learner starts with uninformative standard multivariate normal $N(0, I)$ priors for parameter $\theta_a$ distributions (the Gaussian distribution was chosen due to its conjugate characteristic). At each step it samples from $N(x_t^T\mu_a, \ x_t^T B_a^{-1} x_t)$ for each arm, and plays (greedily) the arm with the highest sampled expected reward. The posterior belief in that arm's $\theta_c$ value is then updated according to the observed context and obtained reward: $B_{a_t} = B_{a_t} + x_t x_t^T$, $f_{a_t} = f_{a_t} + r x_t$, $\mu_{a_t} = B_{a_t}^{-1} f_{a_t}$. The posteriors for the other arms are not updated.

The newly proposed algorithm that exploits the disjoint clustering structure (LinTSC [1]) is based on the same idea as LinTS, but adds an additional level to the selection of which arm to play (it keeps prior beliefs also for each cluster, and updates them according to the obtained rewards). LinTSC relates to LinTS in the same way as TSC relates to TS.

The proposed algorithms based on Thompson sampling are compared to the following ones:

- The LinUCB (Linear Upper Confidence Bound [10]) algorithm solves the CBC problem (it ignores the clustering structure). The arm selection procedure is inspired by its MABC counterpart: the learner plays the arm that maximises the expression: $x_t^T\theta_a \ \alpha\sqrt{x_t^T B_a x_t}$. It basically applies online ridge regression to estimate the parameters. The values for $\theta_a$ and $B_a$ are computed and updated in the same way as in LinTS.

- The LinUCBC (Linear Upper Confidence Bound for Clusters [6]) algorithm is an extension of LinUCB to clustered set of arms. It is based on the LinUCB algorithm, but adds another level to the arm selection procedure: it first selects which cluster and then which arm to play next.

There are no strict assumptions that the synthetic datasets for CBC problems should meet. Contextual data is generated in the same way as in [6]: we have $N$ arms and $K$ clusters, each arm $j$ is uniformly randomly assigned to a cluster $i$. For each cluster we sample a centroid $\theta_i^c \sim N(0, I_5)$ and assign the parameters to its arms as follows: $\theta_j = \theta_i^c + \epsilon\nu$, $\nu \sim N(0, I_5)$. We control the expected diameter of a cluster by varying $\epsilon$. We generate the context at each timestep as is described in [6], sampling them from a multivariate standard normal distribution.

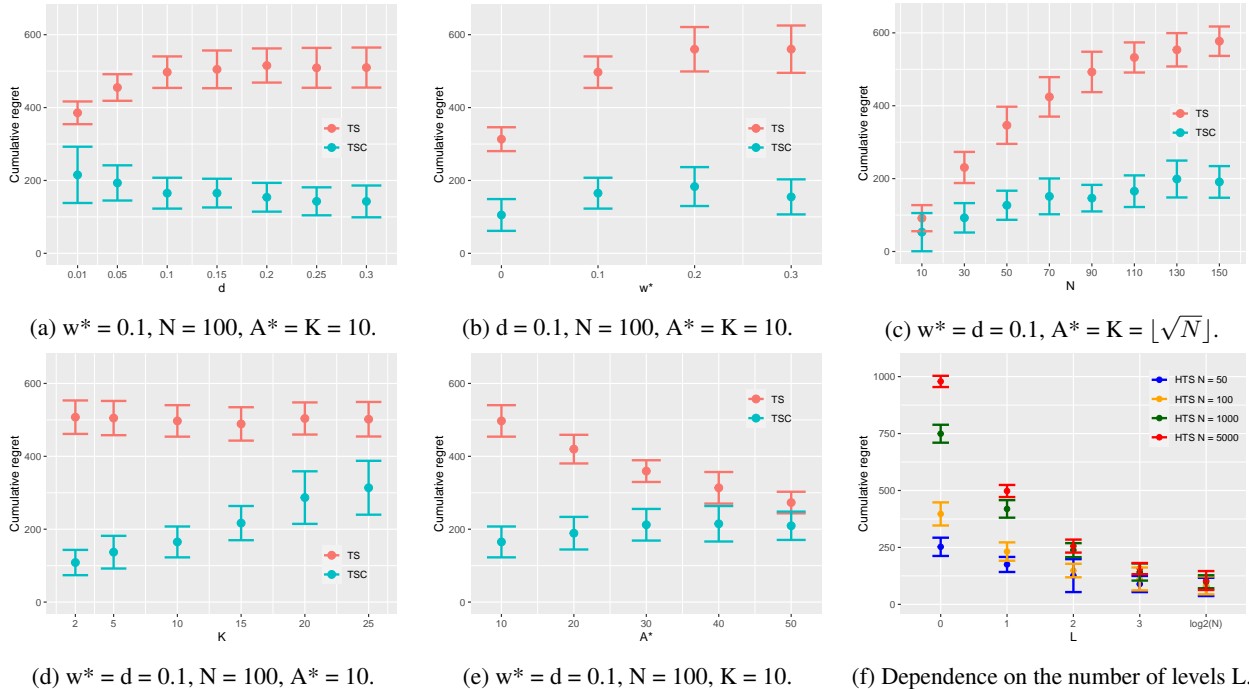

Figure 1: Strong and Hierarchical Strong Dominance. The error bars correspond to ±1 standard deviation.

# 5   Results

There are no exact numbers in the original paper to reproduce, rather the main claims are supported with visualisations. With this in mind, our reproduction confirms the advantage provided by clustering over the assumption of independent arms, as well as the newly proposed algorithms outperforming the referenced benchmarks. We repeat all the experiments with multiple seeds to obtain robust estimates of the algorithms' performance and reduce the risk of drawing any conclusions out of results obtained by chance. We show the obtained results in Figures 1 and 2 (the estimates are obtained with evaluations over multiple – 25 to 100 – random seeds). We provide details on single reproduced claims (as defined in Section 2) in the following subsections.

## 5.1   Classical MABC

In Figure 1 we show all the obtained results from the experiments within the classical MABC problem setting (as described in the previous section), with plots 1a - 1e presenting the disjoint and plot 1f the hierarchical clustering. The results clearly show that taking into account the clustering structure of the arms significantly increases performance (i.e. lowers the cumulative regret). Our results are perfectly in line with those reported in the original paper. In each one of the plots we show how a dataset's hyperparameter affects the learner's performance:

- Figure 1a: an increase in the distance $d$ between the optimal and the other clusters lowers the regret – and its variance – in TSC (it speeds up the recognition of the best cluster), but increases it in TS (most of the arms get a lower expected reward assigned).
- Figure 1b: an increase in the best cluster's width $w$ doesn't affect TSC performance, but negatively affects TS.
- Figure 1c: one of the goals of the original paper was to remove the cumulative regret's dependence on $N$. This plot doesn't show a precisely constant TSC performance, but we can still observe a dramatic improvement over TS.
- Figure 1d: TS doesn't use any clustering information, therefore varying the number of clusters $K$ doesn't affect it at all. On the other hand, increasing $K$ drops TSC performance for the same reason why increasing $N$ negatively affects TS.
- Figure 1e: with more arms in the best cluster, there is a higher chance even for TS to play one of them, therefore we can observe similar performances where half of the arms are grouped into the optimal cluster.

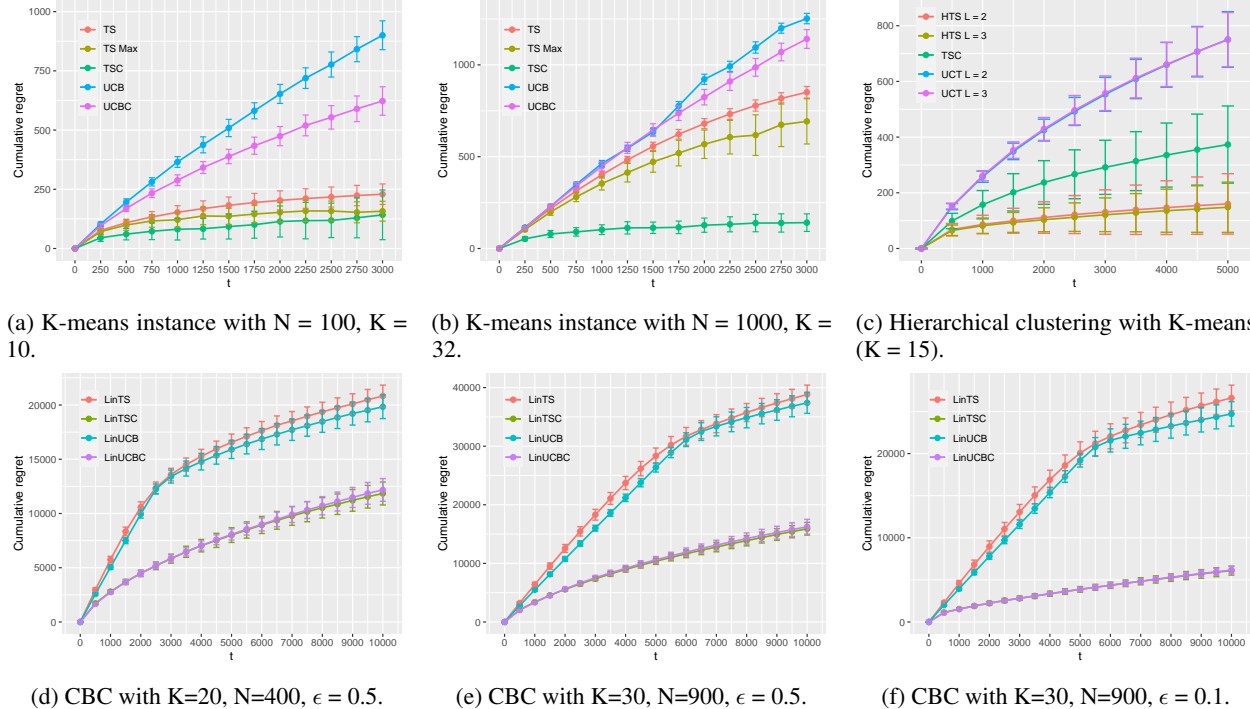

(a) K-means instance with N = 100, K = 10.

(b) K-means instance with N = 1000, K = 32.

(c) Hierarchical clustering with K-means (K = 15).

(d) CBC with K=20, N=400, $\epsilon = 0.5$.

(e) CBC with K=30, N=900, $\epsilon = 0.5$.

(f) CBC with K=30, N=900, $\epsilon = 0.1$.

Figure 2: CBC and violation of assumptions in MABC. The error bars correspond to $\pm 1$ standard deviation.

- Figure 1f: number of arms $N$ effect on HTS learner's performance drops with the number of levels in the hierarchical clustering. Starting with no levels (equivalent to TS), the obtained regret heavily depends on $N$, with its effect on HTS performance dropping to zero in a full binary tree (clustering with $\log_2(N)$ levels).

## 5.2    Classical MABC with violated assumptions

In Figures 2a - 2c we show comparisons of the newly proposed algorithms with some baselines within the classical MABC problem setting with violated assumptions. Here we get some slightly different results than those reported in the original paper:

- Figure 2a: UCB-based algorithms (UCB1 and UCBC) perform much worse that TS-based ones. We observe much better results for TSMax than those reported in the paper. Since TSMax does also exploits clustering and just slightly differs from TSC, we believe the authors must have made some mistakes in its implementation (they report a worse performance than UCBC).

- Figure 2b: TSC clearly stands out from the others in term of performance when we significantly increase the number of arms $N$. Again we observe better TSMax performance than reported.

- Figure 2c: UCT algorithm perform much worse than HTS (note that TSC is an HTS with $L = 1$). The obtained regrets are in line with the original paper, but we observe higher uncertainty in our estimations (although we repeated each experiment the same number of times and show the errorbars in the same way $- \pm 1$ standard deviation).

## 5.3    Contextual CBC

In Figures 2d - 2f we show the results of our experiments with the contextual bandits algorithms. The single plots present the evaluations on datasets generated with different hyperparameters, however they all have the same shape, hence we can analyse all of them together. We can see that in the CBC problem, the algorithms that leverage the clustering information heavily outperform the others, while there is no significant difference between UCB- and TS-based methods. The authors of the original paper here claim that LinTSC slightly outperforms LinUCBC, but from our results we definitely can't draw the same conclusion.

# 6 Discussion

With our work we are able to successfully reproduce the results obtained by the authors of the original paper. As explained in the previous section, some of our results differ slightly from those reported in the paper mostly with respect to variance of the estimates, but they still support the main claims. However, we have to point out a couple of potential issues that we identified in their work.

First of all, Upper Confidence Bound algorithms rely on the initialisation step during which each arm should be played once. If we look again at Figures 2b or 2c, we see that those learners spent a significant amount of the allocated time just playing each arm one by one. Furthermore, for $t < 1000$, the presented numbers are just the result of playing the first $N$ arms, therefore determined by their ordering.

Some more concerns arise when dealing with the CBC problem. In the original paper, each arm has its own parameter vector $\theta_a$ that we want to learn from the obtained rewards and given contexts $x_t$. They mimic the same setting as in Bouneffouf et al. [6]. However, the other two algorithms (LinTS and LinUCB) are designed for a different setting, where a different context is given for each arm at every timestep. Furthermore, per [6], the expected reward should be linearly dependent on the context and played arm's feature vector, but the reward itself should lie inside $[0, 1]$. This is clearly in contrast with our CBC setting, where both negative and larger rewards are possible (and actually really common too). We can't expect dot products of normally sampled vectors to always fulfill these conditions.

## 6.1 What was easy

The authors have included in the paper all the necessary details to reimplement their proposed algorithms, recreate the synthetic datasets, and reproduce the experiments for the first part, i.e. the classical MABC setting.

## 6.2 What was difficult

It is much harder to reimplement some of the referenced benchmarks. The main reasons for these struggles are the inconsistent nomenclature (the referenced paper presents multiple algorithms which were designed with a different name than that one used in the reproduced paper), important details missing in the referenced papers, no code available whatsoever and some of the compared benchmarks being originally designed to run in a different setting (especially true for CBC problems). Furthermore, some additional research into the field of contextual bandits is needed to reproduce the CBC part of the experiments, since due to all the inconsistencies between the different papers, we had a hard time understanding how are contextual bandits supposed to work.

## 6.3 Communication with original authors

There has been no communication neither with the authors of the original article nor with any authors of the referenced papers.

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
