# OpenReview forum: "Thompson Sampling for Bandits with Clustered Arms"
_ML_Reproducibility_Challenge/2021/Fall — RC2021_

### Official Review · Reviewer_N7u8 · 2022-03-01
**great report**

**Rating:** 7
**Confidence:** 4

**Review:**

Reproducibility summary: The major findings are summarized in the summary.

Scope of reproducibility: The scope is clearly stated, but it would be great to include more important information other than the background of the original paper.

Code: The authors implemented everything from scratch.

Communication with original authors: There is no such communication. Since the report points out several potential issues in the original paper, I think it would be great to communicate with the original authors.

Hyperparameter search: More hyperparameters have been experimented.

Ablation study: The experiments on problems with violated assumptions have been conducted.

Discussion on results: The report contains detailed discussion on the empirical results, and even points on the potential issues in the original paper.

Recommendations for reproducibility: The report provides such recommendations.

Results beyond the paper: More empirical results are included in this report.

Overall organization and clarity: The report is clearly written and the overall organization is good.

---

### Official Review · Reviewer_Lm9i · 2022-03-01
**Good reproduction of original paper's code along with the referenced benchmarked code.**

**Rating:** 7
**Confidence:** 3

**Review:**

The paper reproduces the code used in the original paper along with the code for referenced benchmarked results. They made the code publicly available and one can benefit from this work. They validate the claims in the original paper with the empirical results.

The paper probably corrects the plots for TSMax in original paper since as they mention in this paper, TSMax also exploits clustering and hence should enjoy better results. But still the overall claims in the original paper stand correct.

They also point out interesting details with respect to CBC problem about how LinTS and LinUCB are not designed for this setting and how the reward here might not lie in the intended range [0,1].

Eventhough I believe the code is relatively easy to reproduce, it is good that the authors reproduced the work and in turn help the future researchers.

---

### Official Review · Reviewer_nsHJ · 2022-03-01
**Review of [Replication Study:] "Thompson Sampling for Bandits with Clustered Arms"**

**Rating:** 7
**Confidence:** 4

**Review:**

Generally, this report is well-written and clear. It provides a compelling, detailed replication study and convincing results. The accompanying code is clear and well-documented.

Suggestions:
- Since the main quantitative results are about reproducing the figures from the original paper, it would be ideal to either show the figures obtained here side-by-side with the original figures, or, even better, add a plot demonstrating that the results are consistent and where they differ (for example, plot the advantage obtained by the original paper vs. the advantage here)
- More detailed documentation than presently found in your README.md file would be helpful. For instance, if other researchers wanted to modify your code or run it with different examples, it looks like they would need to get the relevant function from `utils.R`. Could you provide high-level documentation of the relevant algorithms/parameters?

Questions:
- Why not obtain, or attempt to obtain, the code from the authors?
- Similarly, it would be helpful to discuss the concerns raised in the Discussion with the authors of the original paper.

Minor points:
- Please clearly identify in the title that this is a study of reproducing the paper.
- P. 3, line -2 "the Thompson sampling" should be "Thomson sampling"
- P. 1 and 3: no need to qualify R as "a well-known language for statistical computing". You can instead cite R - to get the proper format, you can run `citation()` in R.
- P. 3: instead of saying that you "use some of [R]'s basic libraries", you can directly list the libraries used, namely `ggplot2`, `mvtnorm`, and `stats`.
- P. 3 "any GPU" should be "a GPU" (and why would you use a GPU for these tasks? There are only a few R libraries that support GPU usage at all, and they don't appear to be applicable to the tasks studied here)
P. 8 "that one used" should be "the one used"

---

### Meta-Review · Program_Chairs · 2022-04-09

**Recommendation:** Accept
**Confidence:** 5

**Metareview:**

The authors present a well written report.  The report is accepted.

---

### Decision · Program_Chairs · 2022-04-09

**Decision:**

Accept

**Comment:**

Following the recommendation of reviewers and meta-reviewer, the paper is accepted for ML Reproducibility Challenge 2021, and will be published in the upcoming special edition of ReScience Journal.